# Strategies to Increase Prediction Accuracy in Genomic Selection of Complex Traits in Alfalfa (*Medicago sativa* L.)

**DOI:** 10.3390/cells10123372

**Published:** 2021-11-30

**Authors:** Cesar A. Medina, Harpreet Kaur, Ian Ray, Long-Xi Yu

**Affiliations:** 1United States Department of Agriculture-Agricultural Research Service, Plant Germplasm Introduction and Testing Research, Prosser, WA 99350, USA; cesar.medinaculma@wsu.edu; 2Department of Plant and Environmental Sciences, New Mexico State University, Las Cruces, NM 88003, USA; harpr123@nmsu.edu (H.K.); iaray@nmsu.edu (I.R.)

**Keywords:** genomic selection, WGBLUP, *Medicago sativa*

## Abstract

Agronomic traits such as biomass yield and abiotic stress tolerance are genetically complex and challenging to improve through conventional breeding approaches. Genomic selection (GS) is an alternative approach in which genome-wide markers are used to determine the genomic estimated breeding value (GEBV) of individuals in a population. In alfalfa (*Medicago sativa* L.), previous results indicated that low to moderate prediction accuracy values (<70%) were obtained in complex traits, such as yield and abiotic stress resistance. There is a need to increase the prediction value in order to employ GS in breeding programs. In this paper we reviewed different statistic models and their applications in polyploid crops, such as alfalfa and potato. Specifically, we used empirical data affiliated with alfalfa yield under salt stress to investigate approaches that use DNA marker importance values derived from machine learning models, and genome-wide association studies (GWAS) of marker-trait association scores based on different GWASpoly models, in weighted GBLUP analyses. This approach increased prediction accuracies from 50% to more than 80% for alfalfa yield under salt stress. Finally, we expended the weighted GBLUP approach to potato and analyzed 13 phenotypic traits and obtained similar results. This is the first report on alfalfa to use variable importance and GWAS-assisted approaches to increase the prediction accuracy of GS, thus helping to select superior alfalfa lines based on their GEBVs.

## 1. Introduction

Alfalfa (*Medicago sativa* L.) is an autotetraploid (2*n* = 4*x* = 32) perennial forage crop with a genome size of 800–1000 Mb [1]. However, alfalfa breeding is complicated by its high heterozygosity, polysomic inheritance, and out-crossing nature, which hinder the creation of inbred lines. Alfalfa breeding goals target improvement of forage yield, quality, and tolerance to biotic and abiotic stresses. This process requires the selection of perennial plants that can maintain biomass productivity and quality over several years. Therefore, traits must be evaluated over multiple harvests each year for several years. Consequently, genetic gain is slower compared to annual crops. In addition, alfalfa breeding programs have largely focused on recurrent phenotypic selection (PS) in field environments to improve quantitative traits of interest. However, this approach is constrained by breeding population size, genotype × environment interactions, or low heritability of the trait, thus hindering the development of superior varieties.

One promising alternative to recurrent PS is indirect selection based on the use of molecular markers generated, for example, via genotyping by sequencing (GBS) [2]. Markers closely linked to quantitative trait loci (QTL) can then be used for marker-assisted selection (MAS) in breeding programs. Initially, QTLs are detected through genetic mapping or genome-wide association studies (GWAS), where marker-trait associations that exceed specific thresholds are declared statistically significant (Figure 1a). However, MAS is primarily effective for traits controlled by relatively few genes with large effects. For complex traits (e.g., stress tolerance or yield) in elite populations, it can be difficult to clearly identify QTL with major effect because the trait is often controlled by multiple loci possessing small effects.

Soil salinity is one of the major environmental factors limiting the productivity of alfalfa. Genomic tools have been applied to identify important genetic factors that influence salt tolerance in alfalfa. Genetic loci associated with salt stress tolerance have been identified from different studies. Yu et al. (2016) identified 23 markers and 14 functional genes associated with germination under salt stress [3]. Liu et al. (2017) identified 42 markers associated with forage yield, plant height, leaf chlorophyll content, and stomatal conductance under salt stress [4]. Liu et al. (2019) identified 49 markers associated with forage yield, plant height, leaf relative water content, and stomatal conductance under salt stress [5]. Most recently, Medina et al. (2020) identified 27 markers associated with biomass yield and plant vigor under salt stress [6]. Those results highlighted the genetic complexity of salt stress response in alfalfa.

Conventional breeding strategy for improving salt tolerance of alfalfa is time consuming and less effective. Genomic selection (GS) offers the potential to shorten alfalfa breeding and selection cycles. GS is a promising alternative to determine the genetic potential or breeding value of an individual based on whole-genome markers (Figure 1a). This method follows the infinitesimal model, which assumes that a quantitative trait is determined by an infinite number of unlinked and non-epistatic loci, each one with a very small effect that satisfies normality and linearity [7]. This technique uses both parametric and non-parametric statistical models to determine associations of phenotypic trait values with genome-wide molecular markers. This information is subsequently used to predict future breeding values (i.e., genomic-estimated breeding values, GEBVs) for each individual in a population based on their genome-wide marker profile/genotype [8]. Hence, rapid marker-based selection cycles can replace some time-intensive phenotypic selection cycles to accelerate genetic gain. In this paper, we review different GS models and their application to polyploid crops. We also demonstrate the implementation of GS models on a real dataset of alfalfa and potato to identify improved approaches to implement GS in different breeding programs.

## 2. Statistical Methods in GS

There is a wide repertoire of parametric and non-parametric models to obtain GEBVs that differ in complexity, accuracy, and computational requirements (Figure 1b). Some phenotypic traits are highly complex and more difficult to predict using their genetic information, therefore, accuracy in GS modeling is a cornerstone. Model accuracy metric is calculated as the Pearson’s correlation coefficient ($r_{GEBV:y}$) between GEBVs in a training population and observed phenotypes in a testing population. Determining a GEBV can be solved as a regression:(1)y=Xβ+e 
where y is a vector (n×1) of phenotypic outcomes in n observations, X is a matrix (n×p) with p number of markers or predictors in n observations, β is the vector (p×1) of marker effects and e is a vector of residual effects. In GS, however, molecular markers or predictors (*p*) are greater than observations (*n*), generating a large *p* small *n* problem (p≫n). Therefore, estimation of marker effects via multiple regression by ordinary least squares is not possible. To resolve this issue, multiple methods have been developed to handle the high dimensionality of the genomic data. Shrinkage models, such as best linear unbiased prediction using ridge-regression (RRBLUP) [9] or genomic best linear unbiased prediction (GBLUP) [10], are the most popular models used in GS. Both of these models assume that the effect of all single nucleotide polymorphism (SNP) markers is normally distributed with equal variance [9]. Bayesian and least absolute shrinkage and selection operator (LASSO) models assume that some SNPs have large or moderate effects, and others have small or null effects [11]. Finally, machine learning (ML) models like random forest (RF), support vector machine (SVM) and deep learning (DL) algorithms do not assume linearity in the model. ML models can use nonlinear kernels to capture complex SNP–SNP interactions and nonlinear relationships.

### 2.1. Ridge-Regression Best Linear Unbiased Prediction (RRBLUP)

The RRBLUP is a shrinkage method to obtain GEBVs by incorporating genomic information into BLUP using ridge regression (RR). This model has been implemented in the R package RRBLUP [9]. Prediction equations used by RRBLUP assume a priori that all loci explain equal amounts of the genetic variation. The core of the RRBLUP package is the function mixed.solve, which solves any mixed model of the form:(2)y=Xβ+Zu+e 
where y is a vector (n×1) of phenotypic outcomes in n observations, X is a matrix (n×p) with p number of markers or predictors in n observations, β is a vector (p×1) of fixed effects, $u\sim N\left(0,K\sigma_{u}^{2}\right)$ is a vector (n×1) of random effects distributed normally with mean zero and variance $\sigma_{u}^{2}$ and K is a positive semidefinite matrix, Z is a design matrix (n×p) for the random effects, and $e\sim N\left(0,I\sigma_{e}^{2}\right)$ is a vector (n×1) of residual effects distributed normally with mean zero and variance $\sigma_{e}^{2}$ and I is the identity matrix.

### 2.2. Genomic Best Linear Unbiased Prediction (GBLUP)

GBLUP measures the relationship between individuals with the aid of marker data. The difference with RRBLUP is the use of a marker-based relationship matrix named the genomic relationship matrix (GRM) or **G** matrix [12]. The **G** matrix defines the covariance between known relatives in a population, based on DNA marker information. The mixed model for GBLUP analysis uses the following formula:(3)y=1μ+Xβ+Zg+e 
where y, X, β and e were defined in Equation (2), μ is the overall mean, $g\sim N\left(0,G\sigma_{g}^{2}\right)$ is a vector (n×1) of random effects distributed normally with mean zero and variance $\sigma_{g}^{2}$ and G is the G matrix, which can be obtained according to the approach of VanRaden [13]:(4)$$G=\frac{ZZ^{\prime }}{2\Sigma p_{i}\left(1-p_{i}\right)}\;$$
where Z is an identity matrix for the markers and $p_{i}$ is the observed minimum allele frequency (MAF) of all individuals genotyped. However, Yang et al. (2010) combined the information on all SNPs (*i*) coded as 0 = AA, 1 = BB, 2 = AB according to alternative allele dosage to calculate the relationship between individuals *j* and *k* into a GRM ($G_{ijk}$) using a weighting scheme based on allele frequencies:(5)$$G_{jk}=\frac{1}{N}\underset{i}{\sum }G_{ijk}=\{\begin{array}{c} \frac{1}{N}\sum_{i}\frac{\left(w_{ij}-2p_{i}\right)\left(w_{ik}-2p_{i}\right)}{2p_{i}\left(1-p_{i}\right)},j\neq k\;\\ 1+\frac{1}{N}\sum_{i}\frac{w_{ij}^{2}-\left(1+2p_{i}\right)w_{ij}+2p_{i}^{2}}{2p_{i}\left(1-p_{i}\right)},j=k \end{array}\;$$
where $G_{jk}$ is the **G** matrix averaged over all SNP positions in the genome, N is the number of markers, $w_{ij}$ is the element of W pertaining to marker *i* and individual *j*, and $w_{ik}$ is the element of W pertaining to marker *i* and individual *k*. The $G_{ijk}$ or GRM produces the off-diagonal (j≠k) and diagonal (j=k) elements [14]. Based on this approach, Slater et al. (2016) proposed a full autotetraploid model to obtain the G matrix:(6)$$G_{jk}=\{\begin{array}{c} 1+\frac{1}{M}\sum_{i}\frac{\left(w_{ij}-p_{i}\right)\left(w_{ik}-p_{i}\right)}{p_{i}\left(1-p_{i}\right)},j\neq k\;\\ 1+\frac{1}{M}\sum_{i}\frac{w_{ij}^{2}-2p_{i}w_{ij}+p_{i}^{2}}{p_{i}\left(1-p_{i}\right)},j=k \end{array}\;$$
where M is the number of markers × 5 and $p_{i}$ is the frequency of each genotype. The genomic relationship matrices described are based on identity-by-state and simply measure the similarity of alleles between individuals [15].

Analysis results from RRBLUP and GBLUP can be similar; however, GBLUP is more computationally efficient than RRBLUP. GBLUP requires a **G** matrix of dimensions n×n (where n is the number of individuals in the population), whereas RRBLUP requires a genotypic matrix n×m (where m is the number of markers) with high dimensionality. In summary, GBLUP does not provide marker effects but is more time/memory efficient than RRBLUP.

#### Weighted Genomic Best Linear Unbiased Prediction (WGBLUP)

The GBLUP method usually assumes that all SNPs explain the same fraction of genetic variance. However, traits are affected by different genetic architectures associated with SNPs that possess varying effects (e.g., major SNPs). To account for varying effects of different SNP alleles, the weighted GBLUP (WGBLUP) method was developed to incorporate unequal weights for all SNPs [16]. The $G^{*}$ matrix is constructed as follows:(7)$$G^{*}=\frac{ZDZ^{\prime }}{2\Sigma p_{i}\left(1-p_{i}\right)}\;$$
where the asterisk symbol ($G^{*}$) is used to differentiate the weighted G matrix from the regular G matrix, Z is an identity matrix for the markers, D is a diagonal matrix, where each element of the diagonal corresponds to SNP weights, and $p_{i}$ is the observed minimum allele frequency (MAF) of all genotyped individuals. To obtain the D matrix, each element of this matrix is defined as:(8)$$D=\left(\begin{array}{ccc} w_{1} & \cdots & 0\\ \vdots & \ddots & \vdots \\ 0 & \cdots & w_{n} \end{array}\right)$$
where w is weight based on SNP effects from different methods. SNP weights can be obtained from Bayesian Regressions with good results [17]. Other methods to determine weights for SNPs include prioritization based on Wright’s fixation scores ($F_{st}$) [18]. The $F_{st}$ score measures the level of genetic differentiation between populations based on a change in allele frequencies. When $F_{st}$ scores were used to compute relative weights, prediction accuracy increased up to 5%. Ren et al. (2021) developed several methods to optimize WGBLUP by the generation of different weighted **G** matrices. They noted that the choice of an optimal GBLUP matrix will depend on the number of loci controlling the trait. Results indicated that estimated marker-variance-weighted (EVW)-GBLUP was superior for traits controlled by loci of a large effect, and absolute value of the estimated marker-effect-weighted (AEW)-GBLUP was better for traits controlled by loci with moderate effect [19].

### 2.3. Bayesian Models

Bayesian models applied to GS do not assume a normal distribution of marker effects. Instead, they assume that few markers will have large effects on the trait, allowing markers to have different effects and variances. Bayesian models impose stronger shrinkage towards zero on small SNP effects and less shrinkage on relatively large SNP effects. The BGLR R package implements a large collection of Bayesian models [20]. The Bayesian models for continuous variables are represented by the equation:(9)$$y_{i}=1\mu +\overset{m}{\underset{j=1}{\sum }}x_{ij}\beta_{j}+e_{i}$$
where $y_{i}$ is the vector of adjusted phenotypic observations $\left\{y_{1},\ldots ,y_{n}\right\}$, μ is the overall mean for the trait, m is the number of markers or SNPs, $x_{ij}$ is the $i^{th}$ genotype for $j^{th}$ SNP, $\beta_{j}$ is a vector for the effect of the $j^{th}$ SNP, and $e_{i}$ is a vector of residual effects with assumed normal distribution $e\sim N\left(0,I\sigma_{e}^{2}\right)$, where $\sigma_{e}^{2}$ is the residual variance and I is the identity matrix. Bayes A, B, Cπ, and Bayesian LASSO (BL) are the most common models used. All models assume different prior distributions for SNP effects (Table 1).

### 2.4. Machine Learning Models

Machine learning is a field that involves the application of computer algorithms and statistical models to interpret and predict large datasets. Algorithmic modeling is a rapidly developing discipline with strong potential to provide accurate and informative analyses or predictions using large and complex data sets [24]. These models are widely used to solve problems across different disciplines, such as medicine, genomics, natural language processing, and stock market forecasting. Compared with classical statistical models, ML models have fewer assumptions about normality and distribution of data. One important remark is that ML models are being developed much faster than their interpretability, developing a new field to be explored. The most common problem with ML algorithms is data overfitting, which results in models that poorly predict the behavior of future data. To avoid this problem, it is necessary to use a robust validation method, such as cross-validation, which provides an indication of performance on new data. SVM and RF are the most common ML models for classification and regression in GS [25,26].

#### 2.4.1. Support Vector Machine (SVM)

SVM is a machine learning algorithm used in classification or regression problems [27]. The objective of SVM is to find the best hyperplane with the maximal margin in an *n*-dimensional space (genotypic matrix) with respect to a given collection of data (phenotypic values) and predict the correct classification/regression of unseen examples. Support vector regression (SVR) is an application of the SVM. In SVR, each n-dimensional input vector (xi) of p SNP markers is associated with a yi as response variable (e.g., yield), where xi∈ℝp and yi∈ℝ. Then, linear regression f(x) is performed using the following equation [28]:(10)f(x)=w′x+b
where, w is a vector of unknown weights (i.e., regression coefficients) and b is the bias. The training data is used to learn w. The coefficients w and b are estimated by minimizing the following regularized loss function R(C) [10]:(11)R(C)=12‖w2‖+C∑i=1nLε(yi−f(xi))
where ‖w2‖=w’**w**, represents model complexity, and C is a positive cost parameter specified by the user. C determines the trade-off between model complexity and training error, yi−f(xi) is the error associated with ith training data point and Lε is the empirical error measured by ε-intensive loss-function:(12)Lε(yi−f(xi))={0 if |yi−f(xi)|<ε|yi−f(xi)|−ε otherwise
where the loss function is zero (“insensitive”) for any absolute error smaller than a predefined value ε. For an error value larger than ε, the loss function is the difference between the absolute error and ε. ε-SVR solutions is sparseness, with a fraction of errors equal to zero and thereby vanishing in the final model f(x) and only absolute errors > 0 are relevant and used as “support vectors”. f(x) can be assumed as linear or non-linear. For non-linear functions, the data can be mapped into a higher dimensionality space using a kernel space. In nonlinear SVR modeling, different kernels can be used to increase the predictive power of the model. The kernel function provides a solution to the classification/regression dataset by adding an additional dimension to the data. Different kernel functions can be selected to transform input data to feature space. Commonly used kernels in SVM include linear, polynomial, radial basis function (RBF), and sigmoidal kernels (Table 2). In high-dimensional data (i.e., microarrays or GS), the RBF kernel is preferred [28].

#### 2.4.2. Random Forest (RF)

The RF method is a machine learning model for classification and regression problems based on the identification of an objective function and its optimization [29]. The objective function measures the distance between the RF output and desired scores to modify internal parameters to reduce this error. Random forests consist of numerous independent decision trees that are independently trained using a random subset of data. The final prediction is calculated as the average values over all the trees. The RF model attempts to reduce the computational cost to train the model, capture complex interactions and reduce the over-fitting risk in the data [26]. In each decision tree, multiple binary filters are applied to create a bifurcation generating branches and a treelike structure. Every point where the samples are filtered is called a decision node. Optimization in RF consists of determining the best way to split samples at decision nodes based on the predictors. RF can optimize four different hyperparameters to increase the predictive power of the model: total number of observations (*N*), total number of predictors (M), subset of predictors chosen for determining a decision tree (*mtry*), and total number of decision trees to generate the RF (*ntree*). Subsequently, RF creates a series of filters based on the predictor variables. Gini impurity score (as described below) and mean squared error are used to select the best variables in decision nodes.

The RF approach can provide accurate predictions with complex genomic datasets. A very useful feature in the RF model is a function-designated, variable importance metrics, which ranks each SNP’s impact according to the trait. Two approaches to computing variable importance include mean prediction accuracy decrease when a variable/marker is removed, and the mean decrease in impurity (or Gini importance). Gini impurity measures how well a potential split is able to separate the samples of two classes at a particular node. Important limitations when using RF algorithms for GS involve slower processing time when a large number of trees are chosen for the model, or the number of SNP markers is too high. For RF analysis, we observed that a processing limit was reached when the genotypic matrix was composed of more than 10,000 markers.

#### 2.4.3. Deep Learning (DL)

Deep learning is a subfield of machine learning with great success in natural language processing, image recognition, or virtual assistance [30]. The DL architecture uses several layers of nonlinear processing units called hidden layers. Hidden layers allow the network to capture higher-order interactions from the data. This method uses artificial neural network architecture where the perceptron is the fundamental unit for comparison as a neuron in a biological neural network. The implementation of DL in genomic selection is recent, and some studies have reported a modest increase in prediction accuracy in comparison with parametric and non-parametric models [31,32,33,34]. In theory, deep learning could perform better for traits with large epistatic effects and low narrow-sense heritability, a concept which is reinforced by the high predictive ability of mixed models as prediction machinery [35]. Numerically encoded SNPs are the inputs to the first layer to produce a centered vector of phenotypes. The most common DL models used in GS are multilayer perceptron neural network (MLP), convolutional neural network (CNN), and recurrent neural networks (RNN).

The formal description of MLP is a feed-forward DL model composed of multiple perceptrons ordered in hidden layers in a directed graph. In MLP, each layer is fully connected with the next one by nonlinear activation functions, such as rectified linear unit activation function (ReLU), to minimize the mean square error. MLP is flexible because no assumption is made about the joint distribution of inputs and outputs. CNN is a special case of a neural network that uses convolution instead of full matrix multiplication in the hidden layers. The convolution is a function that can be defined as an “integral transform” to reduce the number of hyperparameters to be estimated. CNN was proposed to accommodate situations where input variables are distributed along a space pattern resembling an SNP matrix. CNN seems to perform best in GS because it can detect patterns in the genotypic matrix, discovering correlations between adjacent SNPs. In addition, CNN appears to perform better when epistatic components are important and the narrow-sense heritability is low [34]. It is important to note that DL depends on an adequate hyperparameter choice and high-performance computing with graphics processing units (GPUs) architecture, which can be challenging to implement in small breeding programs.

### 2.5. Other Models

Klápště et al. (2020) presented a strategy to generate a marker-based relationship matrix that prioritized markers using Partial Least Squares (PLS). This approach downweighs noisy predictors, but does not remove them from the model. The advantage of PLS is that it deals with multicollinearity and can handle several response variables at a time. The authors used PLS-Canonical Analysis (PLS-CA) for constructing marker-based relationship matrices with different numbers of markers. This strategy attempts to improve the accuracy of traits with low heritability by taking advantage of the genetic covariance common across all investigated traits. In order to perform the marker selection by PLS-CA, all individuals in the training population must be phenotyped for all traits that will be included in the analysis [36].

The incorporation of staking ensemble ML (SEML) in GS is a promising alternative to increase the predictive ability of ML models. The SEML method uses a meta-learning algorithm to determine how to best combine the predictions from two or more base ML models. Hence, SEML has the potential to generate predictions with better performance than any single model [37]. Liang et al. (2021) tested the prediction accuracy of the SEML approach using three ML models: SVM, kernel ridge regression, and elastic net in Loblolly pine, beef, and dairy cattle. On average, there was an increase of 7.70% in prediction accuracy in nine traits tested. However, Bayes B demonstrated higher prediction accuracies for some traits, including milk fat percentage or tree stiffness [38].

## 3. Genomic Selection in Polyploids

GS requires high-quality genome-wide markers to determine *GEBV*s. Two types of high-throughput genotyping methods can be employed: SNP arrays and GBS. There are SNPs arrays with different marker densities in potato [39] and wheat [40]. Alfalfa also has an array with 9277 SNPs [41]. However, its use has not been widely adopted, and GBS is currently the best option to obtain genome-wide markers. During the genotyping process by GBS, different types of markers, such as single nucleotide polymorphisms (SNPs), insertions/deletions (indels), or short tandem repeats (STRs) can be obtained. Genome-wide markers can then be arrayed in a genotypic matrix of *m* samples and *n* markers. The genotypic matrix can be filtered to retain only biallelic SNPs, which are the most abundant and stable markers for identifying QTLs associated with traits of interest [42].

Allele dosage counts alternative allele frequency for each biallelic SNP. In diploid species the genotypic matrix is coded as {0, 1, 2}, reflecting if a given marker is present in the homozygous reference (AA), heterozygous (AB), or homozygous alternate (BB) allelic state. For biallelic SNPs in polyploid species with ploidy N, the biallelic dosage is N+1 and the genotypic matrix is coded as {0,…,N}. Genotype calling in autotetraploids requires bioinformatics tools to distinguish among five possible genotypes (AAAA, AAAB, AABB, ABBB, BBBB) with biallelic SNPs coded as {0, 1, 2, 3, 4}. There are several R packages, such as polyRAD [43], superMASSA [44], FitTetra 2.0 [45] or Updog [46], with which to obtain allele dosage in numeric format from variant call format file (vcf) format. Some of these R packages, such as Updog, require users to specify genotype priors [46] to accurately calculate the allele dosage and distinguish between all possible genotypes. However, the most common option is to use high depth sequence reads (e.g., ~60×) which leads to 98.4% accuracy in genotypic calls [47] The effects of marker allele dosage on phenotype for genomic selection have been reported previously. Slater et al. (2016) described three different models for GS in autopolyploids: additive autotetraploid, pseudodiploid, and full autotetraploid. In the additive autotetraploid model, the allele dosage has an additive effect, and 0, 1, 2, 3, 4 corresponds to AAAA, AAAB, AABB, ABBB, BBBB, respectively. In the pseudodiploid model, all heterozygous genotypes (AAAB, AABB, ABBB) have the same effect of 1 on the genotypic variation, while the two homozygotes AAAA and BBBB have an effect of 0 and 2, respectively. Finally, the full autotetraploid model assumes that each genotype has its own effect with five possible effects per marker, assuming that the markers are fitted as random effects [15]. In addition, Rosyara et al. (2016) developed GWASpoly, a software for genome-wide association studies in autopolyploids. GWASpoly has different assumptions over allele dosages and conducts the hypothesis tests for each marker using six models (general, diploidized general, diploidized additive, additive, simplex dominant, and duplex dominant models) (Table 3).

Amadeu et al. (2019) evaluated the inclusion of dominance effects for genomic prediction in autotetraploid crops. They reported that a full autotetraploid model, including additive and dominance effects jointly modeled into a unique general effect, increased the total genetic variance explained [48]. In potato, different covariance genomic marker- and pedigree-based matrices, designated **G** and **A**, respectively, were tested to identify additive and nonadditive genetic effects and to improve the accuracy in GS. **A** matrix (also known as numerator relationship matrix) was calculated from a 13-generation pedigree. The **A** matrix was defined as a matrix containing kinship coefficients among all individuals in the population, multiplied by four. They reported that the **G** matrix was superior to the **A** matrix and adding allele dosage information increased the prediction accuracy. Finally, the use of a pseudodiploid matrix reduced the prediction accuracy by 0.13, on average [49].

In the autotetraploid forage grass *Panicum maximum*, de C. Lara et al. (2019) compared the predictive ability of six GS models in six traits using tetraploid and pseudodiploid allele dosages and a minimum depth of 25 reads. Additionally, multiple harvests were modeled with a variance-covariance matrix for genotypes nested across harvests, treating the genotypes as a random factor. The incorporation of correlations among harvests provided a better fit for the traits analyzed. Including the tetraploid dosage also produced higher predictive accuracy compared with pseudodiploid dosages. In autotetraploids with highly mixed ploidy, such as sugarcane and sweet potato, the incorporation of allele dosage information increased model predictive abilities up to 140% in comparison to using diploidized markers [50]. The accuracy of different models showed few changes to ploidy or allele dosage information in sugarcane on sweet potato. In sugarcane, the Brix trait possessed the highest mean predictive accuracy (0.24) using the GBLUP model that included allele dosage. In sweet potato, prediction accuracies were moderate to high. For example, color saturation had the highest mean predictive ability (0.75) using a G model with allele dosage information.

In blueberry (*Vaccinium* spp.), several approaches have been evaluated to improve the genomic selection process, because the conventional breeding pipeline takes up to 12 years [35,51,52]. Implementation of GS in the early stages of the breeding program could shorten the cycle time to nine years and increase the expected genetic gain by 86%. De Bem Oliveira et al. (2019) compared diploid, tetraploid, and continuous allele dosages at the individual plant level for the application of genomic selection in potato and blueberry. In general, there was no difference among the models tested, but continuous genotypes resulted in a better predictive ability for some traits, such as fruit firmness, fruit scar, and fruit diameter. Furthermore, the use of a marker-based relationship matrix generated better predictions than a pedigree-based relationship matrix (**A** matrix). Ferrão et al. (2021) reported similar prediction accuracies of GBLUP for four traits using two genotype calling approaches (dosage and ratio) and two read-depth scenarios (6× and 60×). They also observed that combining allele dosage for low to mid sequencing depths (6×–12×) produced similar accuracies to that obtained by high read-depth (60×). The use of mid sequencing depths will allow modifying economic resource allocation to increase the number of individuals genotyped.

Enormous progress has been made during the last few years in the application of GS approaches to polyploids. In alfalfa, GS has been tested in different traits using parametric and non-parametric models. Table 4 summarizes the progress that has been made towards applying GS in multiple polyploids, including alfalfa. Although yield is the main trait in alfalfa breeding programs, other agronomic traits, such as forage quality and plant regrowth, have also been tested [53]. Furthermore, use of an allele dosage genotype matrix has been reported to improve prediction accuracies of forage quality [54] and yield under salt stress [6]. The current challenge is to implement GS in breeding programs and to evaluate increases in GS-derived genetic gain in comparison with PS-derived materials.

## 4. Case Study: Logan 2020 Population

In this review, we tested some models of GS using the dataset of alfalfa previously published [6]. Datasets were collected from a multi-parental population generated to select lines tolerant to salt stress. Forage yield under salt stress was measured over seven harvests in 265 individuals for two years. Each harvest was spatially corrected by a two-dimensional P-spline mixed model with the Mr.Bean web application [65] using the SpATS package [66]. Multiple best linear unbiased estimator (BLUE) values were adjusted in multi-environmental trials using Factor Analytic II covariance structure [67] with ASReml R software [68]. A genotypic matrix of 6796 high-quality GBS-derived SNP markers was obtained using NGSEP v4.0.0 software [69] with parameters previously reported in [6]. SNPs were coded from 0 to 4 according to allele types using the Sommer R package [70].

A GS approach using regression analysis between phenotypes (y) and a genotypic matrix was transformed by eight different models (f(X)). These included: RRBLUP, Bayes A, B, C, Bayesian Lasso (BL), GBLUP using two G matrices (VanRaden [VR] [Equation (4)] and full-autotetraploid [FA] [Equation (6)]), RF and SVM (Figure 2a). All models were compared for execution time and Pearson’s correlation using ten-fold cross-validation with the GROAN R package [71]. Execution time is an important factor to consider for GS modeling when computing power is limited. Consequently, system time (seconds) was measured for each model with cross-validation. The fastest models were GBLUP and RRBLUP with an average of 0.06 s, whereas SVM and RF required 10.57 and 12.99 s, respectively. More time was required for ML models when a grid search was used to estimate the best values for hyperparameters such as cost and sigma in SVM, or *mtry* in RF. However, time was reduced to 1.90 s with cross-validation in the SVM training model when hyperparameters were previously defined. Prediction accuracy was approximately 0.3 among the GS models RRBLUP, Bayes A, B, C, BL, RF, GBLUP-VR, and GBLUP-FA. The SVM model possessed the highest accuracy (0.46), in agreement with previous reports [6,59].

Variable importance values or SNP weights were obtained using SVM and RF models with the Caret R package [72], or by retrieving −log_10_
*p*-values resulting from six models of the GWASpoly R package (i.e., general, diploidized general, diploidized additive, additive, simplex dominant and duplex dominant models [Table 3]) [73]. SNP weights were used as input in a D diagonal matrix [Equation (8)] for the construction of a $G^{*}$ matrix [Equation (7)] in the WGBLUP model (Figure 2b,c). Pearson’s correlation among variable importance values of different models was measured to identify models with similar SNPs weights. Diploidized additive and diploidized general models had the highest Pearson’s correlation (0.87), followed by additive and diploidized additive models (0.74). Variable importance values derived from RF had low correlations across all models tested (Figure 2c). Prediction accuracies for GBLUP with two **G** matrix and 10 WGBLUP models were compared by measuring Pearson’s correlation 10 times with ten-fold cross-validation. The incorporation of variable importance values in WGBLUP increased prediction accuracies. Pearson correlations ranged from a low of 0.32 in GBLUP-VR (no variable importance values) to 0.63 in WGBLUP-SVM, with the highest prediction accuracy (0.83) achieved when −log_10_
*p*-values from the additive GWASpoly model were used as a weight vector (Figure 2d). Thus, incorporation of a diagonal matrix D with variable importance values to the **G** matrix increased GS predictive ability almost three times without increasing computational time. This is the first report using SNPs weights to increase the prediction accuracy of GS in alfalfa. Our results suggest that including SNP marker −log_10_
*p*-values derived from the additive GWASpoly model in a WGBLUP model may benefit prediction accuracy and selection for improvement of complex traits in alfalfa breeding programs.

## 5. Case Study: Potato Diversity Panel

To ensure that our approach is useful in other polyploid crops, we extended GS analysis in potato. RRBLUP, GBLUP and, WGBLUP models were evaluated in potato using supplemental files from [73]. Phenotypic data consisted of 13 agronomic traits evaluated in 187 lines of the SolCAP potato diversity panel were used. The genotypic matrix was generated by an Infinium SNP array containing 3521 markers with allele dosage. The genotypic matrix was transformed to digital format using the sommer R package [70]. The GBLUP and WGBLUP models were tested using the VanRaden **G** matrix [Equation (4)] and the WGBLUP equation [Equation (7)], respectively. Six different D matrices were generated according to the SNP −log_10_
*p*-values from six models of the GWASpoly (i.e., general, diploidized general, diploidized additive, additive, simplex dominant, and duplex dominant models [Table 3]). G matrices were constructed using the function Gmatrix from the AGHmatrix R package [74]. Prediction accuracies for RRBLUP, GBLUP, and WGBLUP models were compared by measuring Pearson’s correlation 10 times with 10-fold cross-validation using the GROAN R package (Table 5) [71].

Higher prediction accuracies in all 13 agronomic traits of potato were obtained using the WGBLUP model with SNP −log_10_
*p*-values derived from the additive GWASpoly model. Traits of glucose, tuber length, or tuber shape showed accuracies higher than 0.9. It is important to point out that traits of tuber length or tuber shape had high accuracies (0.82 and 0.78 respectively using RRBLUP and GBLUP models) and the use of the WGBLUP model increased the prediction accuracy up to 0.93. Total yield had low prediction accuracies with RRBLUP or GBLUP models (0.132 and 0.117, respectively), and the use of the WGBLUP model increased prediction accuracy by almost five times (Table 5). These results agree with our previous results in alfalfa (Figure 2).

## 6. Conclusions

Genomic selection is a breeding strategy that predicts the genomic estimated breeding value (GEBV) of individuals in a population using genomic-wide genetic markers. A significant advantage of GS is the ability to select superior individuals in a population at very early stages in a breeding cycle based on their genotype, versus conducting lengthy/expensive phenotyping trials prior to each selection cycle. For instance, in a simulated wheat breeding program, selection based on GEBVs for grain yield tripled the genetic gain compared with PS [75]. In alfalfa breeding programs, GS can be implemented in elite large germplasm panels with genotyped individuals to decrease PS efforts, thus reducing overall selection cycle time and accelerating variety development. However, determining correct GEBVs relies on prediction accuracy, which can vary according to a combination of phenotypic trait, genotypic information, and the statistical model. Consequently, GS research efforts have focused heavily on evaluating prediction accuracies of multiple parametric and non-parametric models to develop robust strategies that can be used in testing populations. Once robust modeling strategies are developed, GS has the potential to accumulate thousands of favorable alleles to develop climate-resilient crops with high yield potential. Additionally, as genotyping is required only once for a given population, multiple traits can be associated with the same genotypic matrix to determine GEBVs for each trait, thus making GS a valuable approach in multi-trait selection [76].

There is a need to explore new methodologies to improve molecular and bioinformatic tools for the application of GS in polyploid crops. The development of new approaches to obtain high-quality genome-wide markers will help to resolve the genetic architecture of complex traits. For example, the PRINCESS platform uses long-read sequencing to detect SNPs, indels, or methylation sites with high accuracy [77]. In GWAS and GS, the number of individuals in a population is crucial to maximize statistical power. Therefore, researchers search for genotyping methods that optimize the balance between cost, sample size, and the number of SNPs. In this regard, GBS is a relatively affordable genotyping methodology. However, for polyploid crops, there is a need for high-coverage sequencing (i.e., read-depth) to accurately estimate allele dosage, which increases genotyping cost. Biallelic SNPs are commonly used in polyploid GS because they are the most abundant type and are easier to transform into a numerical format for developing a genotypic matrix. However, during the construction of the vcf file, ~20% of high-quality SNP markers are discarded because they are not biallelic. Additionally, indels or simple sequence repeats (SSR) could add important information to GS model to increase prediction accuracy.

Parametric models such as RRBLUP assume that all SNPs have an effect on a specific trait, but the actual effect of each SNP is very small (heavy shrinkage) [9]. Although multiple loci have an effect on a complex trait, they often have different weights. Thus, identifying trait-specific weights for SNP marker alleles should increase prediction accuracy in GS models such as WGBLUP. Such outcomes have been demonstrated in animal breeding research [78,79], but not in polyploid crops. In this regard, utilizing a variable importance approach based on −log10 *p*-values for the additive GWASpoly model for alfalfa yield under salt stress was the best strategy to generate a diagonal matrix D. For this complex trait, we present empirical evidence demonstrating that the WGBLUP model increased prediction accuracy by almost 3 times compared to RRBLUP, GBLUP, Bayesian or ML models. Finally, we expanded the use of the WGBLUP model to 13 agronomic traits of potato and demonstrated the increase in prediction accuracy up to five times in complex traits, such as yield. The WGBLUP approach is simple, does not require high-performance computing, and can be applied to different crops to predict breeding values and accelerate selection cycles.

## Figures and Tables

**Figure 1 cells-10-03372-f001:**
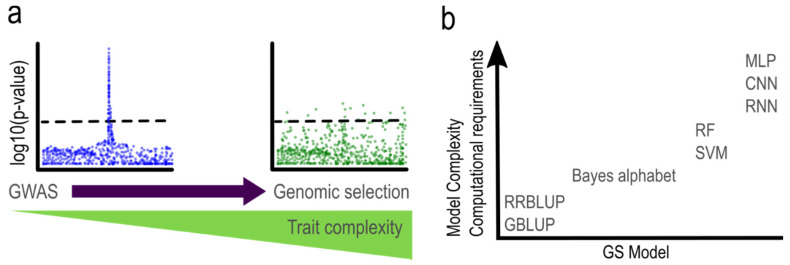
Indirect selection based on molecular markers. (**a**) Generalized Manhattan plots illustrating a comparison of GWAS effectiveness in simple (left) vs. complex traits (right). Note: Bold dashed line indicates minimum threshold to select significant markers. A significant signal (i.e., QTL) was identified in the simple trait (left panel), while no defined QTL was identified for the complex trait. Therefore, genomic selection (GS) is more appropriate and practical for complex traits. (**b**) Common parametric and non-parametric models used in GS and their computational requirements. GBLUP, genomic best linear unbiased prediction; RRBLUP, ridge-regression BLUP; RF, random forest; SVM, support vector machine; MLP, multilayer perceptron; CNN, convolutional neural network; RNN, recurrent neural network.

**Figure 2 cells-10-03372-f002:**
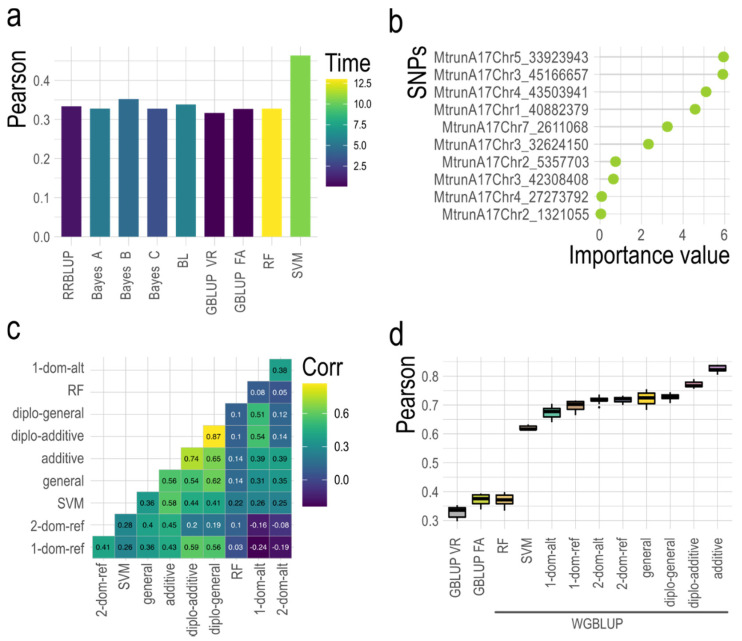
Optimization of GS models. (**a**) GS model accuracy measured as Pearson’s correlation after 10-fold cross-validation for biomass yield under salt stress. Computing time was measured as system time in seconds to run one cross-validation. (**b**) Example of variable importance values derived from SVM for 10 randomly chosen SNPs. (**c**) Pearson’s correlation for 6796 SNPs weights obtained by variable importance (SVM, RF) or by −log_10_
*p*-values of different GWASpoly models. (**d**) Accuracy of GBLUP (GBLUP VR and GBLUP FA) and WGBLUP models. Accuracy was measured 10 times using Pearson’s correlation with 10-fold cross-validation. SNP weights for WGBLUP were obtained from variable importance values (SVM, RF) or −log_10_
*p*-values of different GWASpoly models. RRBLUP, best linear unbiased prediction using ridge-regression; BL Bayes LASSO; GBLUP, genomic best linear unbiased prediction; VR, VanRaden **G** matrix; FA, full autotetraploid **G** matrix; RF, random forest; SVM, support vector machine; WGBLUP, weighted GBLUP; 1-dom-alt and 1-dom-ref, simplex dominant models; 2-dom-alt and 2-dom-ref, duplex dominant models; diplo-general, diploidized general; diplo-additive, diploidized additive.

**Table 1 cells-10-03372-t001:** Different prior distributions for Bayesian models.

Model	Prior Distribution ^‡^	Ref.
Bayes A	$\beta_{j}\sim t\left(df_{\beta },S_{\beta }\right)$	[8]
Bayes B	$\beta_{j}=\{\begin{array}{c} 1/2\gamma \lambda \exp\left(-\lambda \left|\beta_{j}\right|\right)\\ \left(1-\gamma \right) \end{array}\begin{array}{c} for\;\beta_{j}\neq 0\\ for\;\beta_{j}=0 \end{array}$	[21]
Bayes Cπ	$\beta_{j}|\pi ,\;\sigma_{\beta_{j}}^{2}\{\begin{array}{c} \beta_{j}\sim 0\\ \beta_{j}\sim N\left(0,\sigma_{\beta_{j}}^{2}\right) \end{array}\begin{array}{c} with\;prob\;\pi \\ \;\;\;\;\;\;\;\;\;\;\;\;with\;prob\;\left(1-\pi \right) \end{array}$	[22]
Bayesian LASSO	$\beta_{j}\sim DE\left(\lambda^{2},\sigma_{e}^{2}\right)$	[23]

^‡^; $\beta_{j}$, is the additive effect of the $j^{th}$; t, scaled-*t* distribution; $df_{\beta }$, degree of freedom; $S_{\beta }$, scale parameters; γ, fraction of the SNPs that are in linkage disequilibrium with a quantitative trait locus; SNP; π, probability of the marker effect equal to zero; DE, double exponential; λ, parameter of exponential distribution.

**Table 2 cells-10-03372-t002:** Kernels used in support vector machine (SVM) model. Meta-parameters used for tuning include gamma (γ), degree of polynomial (d) and intercept (α).

Kernel	Formula ^‡^
Linear	$K\left(x_{i},y_{j}\right)=x_{i}^{T}y_{j}$
Polynomial	$K\left(x_{i},y_{j}\right)=\gamma {\left(x_{i}^{T}y_{j}+\alpha \right)}^{d}$
Radial basis function	$K\left(x_{i},y_{j}\right)=e^{-\gamma \|x_{i}-y_{j}\|{}^{2}}$
Sigmoidal	$K\left(x_{i},y_{j}\right)=\tan h\left(\gamma x_{i}^{T}y_{j}+\alpha \right)$

^‡^; $x_{i},y_{j}$ are two vectors in the *n*-dimensional space.

**Table 3 cells-10-03372-t003:** Coding effect assumptions of GWASpoly models according to allele dosage in biallelic SNPs.

Allele Dosage ^¶^	AAAA	AAAB	AABB	ABBB	BBBB
Numerical Code	0	1	2	3	4
**GWASpoly Models**	**Phenotypic Effect ^§^**				
Diplo-additive	0.00	0.50			1.00
Diplo-general ^‡^	0.00	0.00 < x <1.00			1.00
Additive	0.00	0.25	0.50	0.75	1.00
1-dom-ref (A > B simplex)	1.00	1.00	1.00	1.00	0.00
2-dom-ref (A > B duplex)	1.00	1.00	1.00	0.00	0.00
1-dom-alt (B > A simplex)	0.00	1.00	1.00	1.00	1.00
2-dom-alt (B > A duplex)	0.00	0.00	1.00	1.00	1.00
General ^†^	No restrictions				

^¶^, allele dosage A is coded as the reference allele and B is coded as the alternative allele; ^§^, phenotypic effects are scaled from 0.00 to 1.00; ^‡^, for the diplo-general model all heterozygotes have the same effect (x), but x is not constrained to be halfway between the homozygous effects; ^†^, the general model has no restrictions on the effects of the different dosage levels.

**Table 4 cells-10-03372-t004:** Recent achievements in genomic selection (GS) in polyploid crops.

Crop	Ploidy	Trait ^§^	GS Method	Acc ^‡^	Notes	Author
*Avena sativa*	Allohexaploid	Seed lipid content	MK-BLUP	0.48	Use of additive marker effects of Bayesian models during the construction of G matrix	[55]
*Brassica napus*	Alloteteraploid	Seed yield	GBLUP	0.69	Several agronomic and seed quality traits were tested	[56]
*Coffea arabica*	Allotetraploid	Canopy diameter	GBLUP	0.40	18 agronomic traits were tested. Diploid dosage assumed	[57]
*Eucalyptus nitens*	Paleotetraploid	Wood density	MVGLUP ^†^	0.77	Marker selection in multivariate analysis. Requires uses multiple traits highly correlated	[36]
*Medicago sativa*	Autotetraploid	Yield	RRBLUP	0.66	Multi-environment trials over two generations. First report of GS in alfalfa.	[58]
*Medicago sativa*	Autotetraploid	Yield	SVM	0.35	Six GS models were tested. First report of machine learning models in alfalfa	[59]
*Medicago sativa*	Autotetraploid	Leaf crude protein	RRBLUP	0.40	Nine alfalfa forage quality traits were tested by five GS models	[54]
*Medicago sativa*	Autotetraploid	Fall plant height	Bayes B	0.65	15 quality traits and 10 agronomic traits were tested using three GS models	[53]
*Medicago sativa*	Autotetraploid	Yield under salt stress	SVM	0.50	Multi-environment trials with seven yield measurements. Eight GS models were tested	[6]
*Panicum maximum*	Autotetraploid	Organic matter	Bayes B-TD	0.39	Genomic selection using tetraploid dosage (GS-TD) vs. diploid dosage (GS-DD)	[60]
*Solanum tuberosum*	Autopolyploid	Yield	GBLUP	0.55	Incorporation of additive and digenic dominant G covariance matrix	[49]
*Solanum tuberosum*	Autopolyploid	Tuber weight	RKHS	0.59	Four agronomic tuber traits were tested by eight GS models	[61]
*Sugarcane*	Octaploid and decaploid	Fiber	GBLUP	0.44	Inclusion of additive and non-additive genetic components for GS	[62]
*Triticum aestivum*	Allohexaploid	Grain yield	GBLUP	0.47	Multi-trait selection for grain yield and protein content	[63]
*Triticum aestivum*	Allohexaploid	Grain yield	GBLUP	0.53	GWAS markers as fixed effects in GS models.	[64]
*Vaccinium corymbosum*	Autotetraploid	Weight	GBLUP	0.49	Comparison of allele dosage with depth sequencing: 6×–60×)	[35]

^§^ For multiple traits, the trait with the highest predictive accuracy was selected; ^‡^, predictive accuracy measured as Pearson’s correlation; MK-BLUP, multi-kernel trait-specific BLUP; MVGLUP, Multi-trait model GBLUP; SVM, support vector machine; Bayes B-TD, Bayes B with tetraploid allele dosage; RKHS, Reproducing Kernel Hilbert Space; ^†^ In multi-trait genomic selection (MT-GS) a secondary trait that is genetically correlated with the primary trait is incorporated in the prediction model, to predict the primary trait with higher accuracy.

**Table 5 cells-10-03372-t005:** Comparison of genomic selection (GS) models in 13 phenotypic traits collected in the SolCAP potato diversity panel. Mean and standard deviation of Pearson’s correlation obtained by 10-fold cross validation in 10 replicates. SNP weights for WGBLUP were obtained from −log_10_
*p*-values of different GWASpoly models.

Trait	RRBLUP	GBLUP	WGBLUP							
			1-d-a	1-d-r	2-d-a	2-d-r	General	d-Gen	d-Add	Additive
Chip color	0.723	0.721	0.826	0.798	0.859	0.850	0.867	0.849	0.855	0.896
	(±0.014)	(±0.015)	(±0.009)	(±0.011)	(±0.007)	(±0.013)	(±0.008)	(±0.009)	(±0.007)	(±0.007)
log_10_ fructose	0.682	0.676	0.819	0.785	0.845	0.833	0.868	0.839	0.855	0.895
	(±0.024)	(±0.025)	(±0.014)	(±0.017)	(±0.007)	(±0.011)	(±0.011)	(±0.015)	(±0.003)	(±0.008)
log_10_ glucose	0.678	0.668	0.796	0.809	0.855	0.849	0.875	0.844	0.848	0.91
	(±0.017)	(±0.030)	(±0.009)	(±0.016)	(±0.009)	(±0.009)	(±0.009)	(±0.011)	(±0.013)	(±0.007)
Malic acid	0.602	0.598	0.751	0.745	0.802	0.801	0.838	0.808	0.826	0.876
	(±0.016)	(±0.027)	(±0.021)	(±0.022)	(±0.021)	(±0.016)	(±0.011)	(±0.016)	(±0.009)	(±0.007)
Sucrose	0.539	0.519	0.676	0.675	0.702	0.716	0.725	0.722	0.739	0.806
	(±0.024)	(±0.034)	(±0.011)	(±0.022)	(±0.019)	(±0.015)	(±0.023)	(±0.011)	(±0.019)	(±0.011)
Total yield	0.132	0.117	0.401	0.413	0.418	0.428	0.470	0.492	0.504	0.584
	(±0.023)	(±0.041)	(±0.026)	(±0.030)	(±0.031)	(±0.017)	(±0.029)	(±0.030)	(±0.030)	(±0.028)
Tuber eye depth	0.495	0.478	0.605	0.655	0.693	0.717	0.740	0.693	0.736	0.812
	(±0.026)	(±0.019)	(±0.029)	(±0.016)	(±0.025)	(±0.014)	(±0.020)	(±0.020)	(±0.018)	(±0.007)
Tuber length	0.826	0.821	0.891	0.884	0.899	0.889	0.904	0.908	0.912	0.928
	(±0.012)	(±0.014)	(±0.006)	(±0.009)	(±0.006)	(±0.012)	(±0.008)	(±0.008)	(±0.005)	(±0.009)
Tuber shape	0.775	0.780	0.865	0.853	0.886	0.863	0.896	0.89	0.891	0.922
	(±0.018)	(±0.017)	(±0.010)	(±0.013)	(±0.008)	(±0.005)	(±0.010)	(±0.008)	(±0.009)	(±0.006)
Tuber size	0.501	0.499	0.641	0.650	0.679	0.663	0.666	0.661	0.679	0.742
	(±0.024)	(±0.027)	(±0.019)	(±0.020)	(±0.020)	(±0.022)	(±0.024)	(±0.022)	(±0.019)	(±0.021)
Tuber width	0.635	0.638	0.752	0.749	0.782	0.772	0.805	0.789	0.803	0.847
	(±0.023)	(±0.021)	(±0.020)	(±0.021)	(±0.016)	(±0.018)	(±0.012)	(±0.015)	(±0.013)	(±0.017)
Vine maturity 95 days	0.288	0.286	0.550	0.538	0.603	0.589	0.668	0.632	0.65	0.746
	(±0.035)	(±0.042)	(±0.028)	(±0.020)	(±0.022)	(±0.028)	(±0.022)	(±0.019)	(±0.025)	(±0.017)
Vine maturity 120 days	0.321	0.323	0.495	0.569	0.636	0.633	0.669	0.616	0.666	0.755
	(±0.047)	(±0.024)	(±0.026)	(±0.021)	(±0.021)	(±0.013)	(±0.025)	(±0.023)	(±0.026)	(±0.019)

RRBLUP, best linear unbiased prediction using ridge-regression; GBLUP, genomic best linear unbiased prediction using VanRaden **G** matrix; WGBLUP, weighted GBLUP; 1-d-a and 1-d-r, simplex dominant models; 2-d-a and 2-d-r, duplex dominant models; d-gen, diploidized general; d-add, diploidized additive.

## Data Availability

Alfalfa raw data of GBS were submitted to the NCBI Sequence Read Archive with bioproject ID: PRJNA611554 and biosample # SAMN14336867 and phenotypic data can be retrieved from [6] (Table S2). Genotypic and phenotypic potato data were obtained from SolCAP potato diversity panel and have been published in [73] as Table S1.csv and Table S2.csv.

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
