# Peer review of "Strategies to Increase Prediction Accuracy in Genomic Selection of Complex Traits in Alfalfa (Medicago sativa L.)"

_cells, 2021, doi:10.3390/cells10123372_

Round 1
Reviewer 1 Report
Not much to say about this MS. It is well-prepared, well-written and failrly comprehensive. Requires minimal or no editing or revision. Nice work.
Author Response
Reviewer 1
Not much to say about this MS. It is well-prepared, well-written and fairly comprehensive. Requires minimal or no editing or revision. Nice work.
Thank you
Reviewer 2 Report
My comments are in the attached pdf file.

Author Response
Reviewer 2
My comments are in the attached pdf file.
by "conventional strategies can be replaced" with "through conventional breeding approaches"
The change has been applied in abstract line 15
previous results? reference?
The previous results and references are shown in table 4
First, can write the technical name and after that can use general name.
The change has been applied in line 18
can named the other polyploids too
We added potato.
How about the other polyploid crops?
In this version of the paper, we extended the WGBLUP model to potato using supplemental files from Rosyara et al. (2016). We include it in section 5 Case study: Potato Diversity Panel. In table 5 we show that the WGBLUP increased the prediction accuracy up to 0.922 in Tuber shape.
Author can talk about the background of biomass yield and salt stress related QTLs studied in polyploids including Alfalfa. What is current status?
In this version of the paper, we include the background and current status of biomass under salt in the introduction: lines 56-65.
Can write the technical name
The change has been applied in line37
Reviewer 3 Report
This is a high quality review.
Author Response
Reviewer 3
This is a high quality review.
Thank you
Reviewer 4 Report
The manuscript entitled “Strategies to increase prediction accuracy in genomic selection of complex traits in Alfalfa (Medicago sativa L.)” is in the scope of Cell journal. The manuscript provides valuable information on employing genome-wide association studies to increase the prediction accuracy of genomic selection in alfalfa and other polyploid crops. Thus I suggest accepting the manuscript following major revision. The manuscript needs major English editing. The presented models of genomic selection need to be improved. The approaches to improve prediction accuracy in genomic selection need to be more highlighted and improved. The implemented models on real data of alfalfa need to be extended.
Author Response
Reviewer 4
The manuscript entitled “Strategies to increase prediction accuracy in genomic selection of complex traits in Alfalfa (Medicago sativa L.)” is in the scope of Cell journal. The manuscript provides valuable information on employing genome-wide association studies to increase the prediction accuracy of genomic selection in alfalfa and other polyploid crops. Thus I suggest accepting the manuscript following major revision.
The manuscript needs major English editing.
The English have been checked and corrected.
The presented models of genomic selection need to be improved.
We believe that the models have been significantly improved in this review paper. the use of WGBLUP increased twice the accuracy predictions in alfalfa and almost five times in potato.
The approaches to improve prediction accuracy in genomic selection need to be more highlighted and improved.
We described in detail the methods and models to improve the prediction accuracy in GS through sections 2, 4, and 5.
The implemented models on real data of alfalfa need to be extended.
We extended our approach to potato using supplemental files from Rosyara et al. (2016). We include in section 5 Case study: Potato Diversity Panel. The accuracy results were included in table 5 where WGBLUP increased the prediction accuracy up to 0.922 in Tuber shape. The results in potato are consistent with that in alfalfa.
Round 2
Reviewer 2 Report
The authors have addressed my raised concerns.
Reviewer 4 Report
The manuscript entitled “Strategies to increase prediction accuracy in genomic selection of complex traits in Alfalfa (Medicago sativa L.)” has been significantly improved in the revised version and all previous comments have been addressed.